# Rediscovering the Shift-Share EM2 Model: A Decomposition Framework of Unbalanced Employment Growth at the Industrial Level

**Jian Luo [1,2,\*]** and **Yongchun Yang [1,3]**

1   School of Earth and Environmental Sciences, Lanzhou University, Lanzhou 730000, China
2   School of Resource and Environment, Xichang University, Xichang 615013, China
3   Key Laboratory of Western China's Environmental Systems, Ministry of Education of the People's Republic of China, Lanzhou University, Lanzhou 730000, China
\*   Correspondence: aqualuo1985@163.com

**Abstract:** The unbalanced growth of employment among regions is ultimately attributed to the difference in industrial structure and industrial competitiveness. At the regional level, both shift-share EM1 and EM2 models can realize the purpose of the shift-share model, that is, thoroughly separate the effects of the industrial structure and industrial competitiveness on employment growth deviation. While the expectation is to further achieve a complete separation on employment growth deviation at the industrial level, the EM1 model is not competent, and this is precisely the role of the EM2 model—to separate the employment growth deviation caused by the differences in industrial structure and industrial competitiveness at the industrial level. Therefore, the shift-share EM2 model can realize the separation of the structural component and competitive component at the industrial level. In the process of industrialization, the manufacturing industry presents a significant trend of unbalanced growth in both spatial and industrial dimensions. The setting of the share component in the EM2 model covers the balanced growth of employment in both spatial and industrial dimensions. The allocation component is reinterpreted as the magnification or inhibition effect of industrial competitiveness on the initial industrial structure. In order to exclude the influence of the regional employment scale difference, the concept and calculation method of the regional deviation degree is put forward. Finally, the spatial pattern of unbalanced employment growth in China's manufacturing industry from 1999 to 2019 was analyzed using the shift-share EM2 model at the regional level. The results show that the spatial imbalance trend of employment growth in China's manufacturing industry is significant, the regional deviation scale of employment growth forms a north–south difference pattern, and the regional deviation degree forms a southeast to northwest difference pattern. Taking Sichuan Province as an example, the application of the shift-share EM2 model at the industrial level is demonstrated.

**Keywords:** shift-share model; employment growth; allocation component; China; manufacturing



## 1. Introduction

In the first half of the 20th century, continuously increasing productivity accelerated the division of production between regions. Then, the expansion of industrial specialization on the spatial scale led to the further concentration of production, capital, and employment, thus aggravating the unbalanced trend of economic development in capitalist countries. The unbalanced growth of regional employment dominated by the industrial structure and industrial competitiveness has gradually become the focus of scholars' research. Therefore, the shift-share model has been applied as an effective decomposition tool for analyzing regional employment growth [1]. The basic methodology of shift-share analysis is deviation analysis and the accounting decomposition structure. First, the employment growth is decomposed into the expected increment under the assumption of balanced growth (share

component) and the growth deviation between the actual increment and the expected increment (shift component). Secondly, the shift component is further decomposed into the structural component and competitive component, which represent unbalanced growth caused by the regional industrial structure and competitiveness difference. The basic idea and methodology of the shift-share model can be traced back to the study of the geographical distribution of the industrial population of Great Britain in the Barrow Report in 1940 [2]. This study clearly defined what "fair share" and "shift" mean. It should be pointed out that Jones noted "analysis of the statistical evidence prepared by MacDougall" at the end of the memorandum [2]. This was also confirmed by MacDougall, who pointed out that the memorandum developed techniques that had not been used before and which have since been used a good deal by students of regional economics [3]. Creamer used the shift-share approach to analyze the manufacturing industry of the United States, but he gave no source for his methodology [4]. The widespread application of the shift-share model is attributed to Dunn's introduction in the annual report of the American Association for Regional Science in 1958. He claimed that the idea for this shift computation was based upon a method formerly used by Creamer [5]. Dunn's contribution was to further decompose "shift" into "differential shift" (competitive component) and "proportionality shift" (structural component). Many important extensions continue to emerge during model evolution. Arcelus proposed the regional effects to evaluate regional factors that account for the strength of the local market [6]. To take into account the continuous changes in both industrial mix and size of employment, Barff proposed a dynamic shift-share analysis, which calculated all components on an annual basis and then summed the results over the study period [7]. Ray used the standardized growth rate to separate the region effect and industry-mix effect [8]. Spatial shift-share models incorporated the spatial structure within shift-share analysis by defining the neighborhood and spatial weight matrix [9–12]. Artige observed a difference between the regional and national aggregate employment growth rates and separated the industry mix effect from the competitive effect [13]. Ruault generalized the shift-share analysis to encompass the spatial concentration of industry and industry emergence and catastrophic growth events based on the model of the Artige [14].

The difference in the shift component among regions is attributed to the difference in industrial structure and industrial growth rate (the positive deviation effect exists in regions with a specialization advantage in fast-growing sectors). Due to the different initial industrial structure, the final employment scale of the two regions is still different even if the overall employment scale is the same and the employment amount of all industries increases at the same rate [15]. The purpose of further decomposition of the shift component is to calculate the deviation caused by the difference in industrial structure and growth rate, respectively. However, Dunn's model does not achieve this goal, so the competitive component and structural component are intertwined. To solve this problem, Esteban-Marquillas derived two shift-share models (hereinafter referred to as EM1 and EM2) by embedding "homothetic employment" [16]. The EM1 model was quickly accepted by scholars and applied to regional employment and economic analysis. Surprisingly, the EM2 model is rarely adopted by scholars, probably because Keil pointed out that the national growth effect and industry mix effect are identical to their classic shift-share analogs when aggregated over industries, so the two models are identical at the regional level [17]. In fact, the share component and structural component of the two models are distributed differently across industries. Loveridge acknowledges the separation effect of the EM2 model but believes that the EM2 model has no practical reference value [18]. Precisely, the cognition of the EM2 model requires reinterpretation to the allocation component and to figure out its relation to the net shift component. The essence of the shift-share model is to decompose the difference of employment growth, rather than the composition of employment growth. The share component of the EM2 model sets the same initial industrial structure and average industrial growth rate for balanced employment growth. However, the share component of the EM1 model only includes the same industrial growth rate for employment balance growth. The unbalanced growth of employment among industries is ultimately attributed

to the difference in initial industrial structure and industrial growth rate. If we expect to separate the structural component and competitive component at the industrial level, the EM1 model is not competent, and this is precisely the role of the EM2 model—to separate the employment growth deviation caused by the difference in initial structure of industry and industrial competitiveness at the industrial level.

Industrial agglomeration is the result of differences in industrial specialization and competitiveness. The systematic research on industrial agglomeration started from Marshall's industrial district theory, and he believed that it was the external economy composed of the specialized input, centralized job market, and knowledge spillover that led to industrial spatial agglomeration [19]. The new industrial district theory, which takes "Third Italy", a social regional production complex dominated by small and medium-sized enterprises, as the research object, holds that the organization of external transaction costs based on the competitive cooperative relationship enterprise network is the basis of industrial agglomeration [20,21]. The new economic geography assumes increasing returns to scale and imperfect competitive markets, and holds that increasing returns to scale and reducing transport costs and the flow of production factors promote industrial agglomeration through market effects [22]. The regional innovation system theory emphasizes the importance of a sociocultural background and a spatial proximity to high-tech industry agglomeration [23]. According to the social capital theory, economic behavior is embedded in the social relationship network and has the embeddedness to deviate from the goal of benefit maximization [24]. Industrial agglomeration continues to strengthen the trend of unbalanced growth of regional employment.

Esteban-Marquillas's description of the allocation component only reveals the distribution characteristic of the regional industrial structure with a relative competitive advantage. However, it does not directly explain the meaning of the allocation component. In addition, due to the difference in employment scale, the degree of regional deviation cannot be directly compared with the regional shift component.

In the following, we first sort out the main shift-share expansion models, which embedded the standard structure in the two basic models. Second, the differences between the shift-share EM1 and EM2 models are compared. Then, the separation effect of the structural component and competitive component in the shift-share EM2 model is verified, and the definition of the allocation component is reinterpreted. Finally, the shift-share EM2 model is applied in analyzing regional unbalanced employment growth by using manufacturing employment data from China in 1999 and 2019.

## 2. Basic Models and Extensions

### 2.1. Two Basic Models

Although there are many extension models for shift-share analysis, most of them are developed on the basis of the classical model and structural base model. The classical model of shift-share analysis was proposed by Dunn, also known as the NGR (national growth rate) model:

$$\sum_{i=1}^{m} X_{ij}R_{ij} = \sum_{i=1}^{m} X_{ij}R + \sum_{i=1}^{m} X_{ij}(R_i - R) + \sum_{i=1}^{m} X_{ij}(R_{ij} - R_i) \tag{1}$$

where $X$ represents the amount of employment (or output value, income, number of companies, etc.); $R$ is the growth rate; subscript $i$ represents industries and sets the number of industries to $m$; subscript $j$ represents regions; the number of regions is set as $n$. The left side of the equation is the regional employment increment during the study period. On the right are the share component (also known as the national growth effect), the structural component (also known as the industry mix effect), and the competitive component. In the shift-share model, the share component represents the expected increment in the regional employment based on the national average level. The shift component (the sum of the structural component and competitive component) represents the deviation in the

actual increment relative to the expected increment. In the shift component, the structural component represents the effect of the industrial structure difference on employment growth deviation, and the competitive component represents the effect of the industrial competitiveness difference on employment growth deviation.

Thirlwall aims to decompose regional employment growth differences and put forward the embryonic form of the structural foundation model [25]:

$$R_j - R = \sum_{i=1}^{m} \frac{X_{ij}}{X_j} R_{ij} - \sum_{i=1}^{m} \frac{X_i}{X} R_i = \sum_{i=1}^{m} \left( \frac{X_{ij}}{X_j} - \frac{X_i}{X} \right) R_i + \sum_{i=1}^{m} \frac{X_{ij}}{X_j} (R_{ij} - R_i) \quad (2)$$

After the transformation of the above equation (multiplying both sides of the equation by $X_j$ and then transposing it), it becomes the structural base model:

$$\sum_{i=1}^{s} X_{ij} R_{ij} = \sum_{i=1}^{s} X'_{ij} R_i + \sum_{i=1}^{s} (X_{ij} - X'_{ij}) R_i + \sum_{i=1}^{s} X_{ij} (R_{ij} - R_i) \quad (3)$$

In this formula, $X'_{ij}$ is named the "standard structure" by Bishop. That means that each industry redistributes regional employment in proportion to the national industrial structure in the base period [26]:

$$X'_{ij} = X_j \frac{X_i}{X} \quad (4)$$

### 2.2. Model Extensions Based on the Standard Structure

Bishop introduced the standard structure into the structural component of the classical model to form the synthesis model [26]:

$$\sum_{i=1}^{m} X_{ij} R_{ij} = \sum_{i=1}^{m} X_{ij} R + \sum_{i=1}^{m} X'_{ij} (R_i - R) + \sum_{i=1}^{m} (X_{ij} - X'_{ij})(R_i - R) + \sum_{i=1}^{m} X_{ij} (R_{ij} - R_i) \quad (5)$$

Arcelus introduced the standard structure into all components of the classical model [6], forming the model as follows:

$$\begin{aligned} \sum_{i=1}^{m} X_{ij} R_{ij} = &\sum_{i=1}^{m} X'_{ij} R + \sum_{i=1}^{m} (X_{ij} - X'_{ij}) R + \sum_{i=1}^{m} X'_{ij} (R_i - R) + \sum_{i=1}^{m} (X_{ij} - X'_{ij})(R_i - R) \\ &+ \sum_{i=1}^{m} X'_{ij} (R_{ij} - R_i) + \sum_{i=1}^{m} (X_{ij} - X'_{ij})(R_{ij} - R_i) \end{aligned} \quad (6)$$

Based on the classical model, Esteban-Marquillas introduced the standard structure (which he called homothetic employment) into the competitive component [16], resulting in the EM1 model as follows:

$$\sum_{i=1}^{m} X_{ij} R_{ij} = \sum_{i=1}^{m} X_{ij} R + \sum_{i=1}^{m} X_{ij} (R_i - R) + \sum_{i=1}^{m} X'_{ij} (R_{ij} - R_i) + \sum_{i=1}^{m} (X_{ij} - X'_{ij})(R_{ij} - R_i) \quad (7)$$

At the same time, Esteban-Marquillas introduced the standard structure into the competitive components of the structure base model, resulting in the EM2 model as follows:

$$\sum_{i=1}^{m} X_{ij} R_{ij} = \sum_{i=1}^{m} X'_{ij} R_i + \sum_{i=1}^{m} (X_{ij} - X'_{ij}) R_i + \sum_{i=1}^{m} X'_{ij} (R_{ij} - R_i) + \sum_{i=1}^{m} (X_{ij} - X'_{ij})(R_{ij} - R_i) \quad (8)$$

### 3. Model Comparison and Validation of the Separation Effect and Reinterpretation of the Allocation Component

*3.1. Comparison between the EM1 and EM2 Models*

The shift-share EM1 and EM2 models have the same competitive and allocation components. Therefore, the difference between the two models lies in the share component and the structural component.

Because:

$$\sum_{i=1}^{m} \frac{X_i}{X} R_i = R \tag{9}$$

$$\sum_{i=1}^{m} X'_{ij} R_i = \sum_{i=1}^{m} X_i \frac{X_j}{X} R_i = \sum_{i=1}^{m} X_j \frac{X_i}{X} R_i = X_j R = \sum_{i=1}^{m} X_{ij} R \tag{10}$$

Therefore, the share components calculated at the regional level of the EM1 and EM2 models are equal. Based on this, it can be further deduced that the structural components of the EM1 and EM2 models calculated at the regional level are also identical:

$$\sum_{i=1}^{m} \left( X_{ij} - X'_{ij} \right) R_i = \sum_{i=1}^{m} X_{ij} R_i - \sum_{i=1}^{m} X'_{ij} R_i = \sum_{i=1}^{m} X_{ij} R_i - \sum_{i=1}^{m} X_{ij} R = \sum_{i=1}^{m} X_{ij} (R_i - R) \tag{11}$$

However, the share component and structural component of the EM1 and EM2 models have different distribution values among industries. To describe this phenomenon intuitively, we design a set of hypothetical data to illustrate it. In this hypothetical circumstance, region A is the benchmark region, and the remaining three regions are set to be different from region A in one or two aspects of industrial structure and industrial growth rate. As seen in Table 1, the calculation results of hypothetical data confirm our previous inference: the sum of the share components and structural components of the EM1 and EM2 models is equal at the regional level (Columns 6 and 8, and 10 and 12 of Table 1, respectively), but the values of the distribution in each industry are different (Columns 5 and 7, and 9 and 11 of Table 1, respectively).

**Table 1.** The difference verification between the share component and structural component of the shift-share EM1 and EM2 models based on hypothetical data.

| Regions | Industries | $X_{ij}$ | $R_{ij}$ | $X'_{ij}R_i$ | $\sum_{i=1}^{s} X'_{ij}R_i$ | $X_{ij}R$ | $\sum_{i=1}^{s} X_{ij}R$ | $\left( X_{ij} - X'_{ij} \right) R_i$ | $\sum_{i=1}^{s} \left( X_{ij} - X'_{ij} \right) R_i$ | $X_{ij}(R_i - R)$ | $\sum_{i=1}^{s} X_{ij}(R_i - R)$ |
|---|---|---|---|---|---|---|---|---|---|---|---|
| A | a1 | 200 | 0.2 | 72.50 | | 82.50 | | −19.77 | | −29.77 | |
| | a2 | 500 | 0.5 | 205.00 | 412.50 | 206.25 | 412.50 | 68.33 | 29.27 | 67.08 | 29.27 |
| | a3 | 300 | 0.3 | 135.00 | | 123.75 | | −19.29 | | −8.04 | |
| B | b1 | 500 | 0.3 | 72.50 | | 206.25 | | 59.32 | | −74.43 | |
| | b2 | 200 | 0.6 | 205.00 | 412.50 | 82.50 | 412.50 | −95.67 | −55.63 | 26.83 | −55.63 |
| | b3 | 300 | 0.5 | 135.00 | | 123.75 | | −19.29 | | −8.04 | |
| C | c1 | 200 | 0.2 | 72.50 | | 82.50 | | −19.77 | | −29.77 | |
| | c2 | 300 | 0.5 | 205.00 | 412.50 | 123.75 | 412.50 | −41.00 | −2.92 | 40.25 | −2.92 |
| | c3 | 500 | 0.3 | 135.00 | | 206.25 | | 57.86 | | −13.39 | |
| D | d1 | 200 | 0.3 | 72.50 | | 82.50 | | −19.77 | | −29.77 | |
| | d2 | 500 | 0.6 | 205.00 | 412.50 | 206.25 | 412.50 | 68.33 | 29.27 | 67.08 | 29.27 |
| | d3 | 300 | 0.5 | 135.00 | | 123.75 | | −19.29 | | −8.04 | |

The share component of the shift-share model represents the balanced growth of regional employment. The share component of the EM2 model sets the balanced growth situation as follows: all regions have the same industrial structure in the base period, and the same industry has the same growth rate. Table 1 shows that in share component of the EM2 model, all regions adopt the same industrial distribution (72.50:205.00:135.00), excluding differences in industrial structure and growth rate. Regions with the same overall employment scale have the same share component with the same industrial distribution. The share component of the EM1 model sets balanced growth as follows: all industries and regions grow at the same rate, excluding only the difference in growth rate, which means

that the industrial structures of all regions remain unchanged and different from each other in the growth process.

At the regional level, the structural component of the shift-share model refers to the fact that if more employment is concentrated in the relatively fast-growing industrial sector, the structural component will be larger, and the specialization advantage will be more pronounced. In Table 1, the average growth rates of the three industries in regions A, B, C, and D are 0.2636, 0.5467, and 0.3857. Although the overall employment scale of the four regions is the same, half of total employment of regions A and D is concentrated in the fastest-growing secondary industry. Hence, regions A and D have the same positive structural component. Half of the total employment in region B and C is concentrated in the primary and tertiary industries with relatively slow growth rates. Hence, both the regions B and C have a negative structural component. The structural components of the shift-share EM1 and EM2 models at the regional level are the same, and the difference is reflected in the industrial distribution of the structural components. Obviously, it is necessary to determine which industries derive the structural deviation of a region. At the industrial level, the structural component of the shift-share EM2 model refers to the employment growth deviation caused by the difference in the proportion of industry in the base period under the assumption of the same industrial growth rate. In Table 1, the average structure of the three industries in base periods of regions A, B, C, and D is 27.50%, 37.50%, and 35.00%. The b1 industry owns the maximum employment scale in the base period of all regions, and its structural component is 59.32 in the EM2 model but $-74.43$ in the EM1 model. The structural component value of b1 in the EM1 model is negative, which cannot reflect the specialization advantage of this industry in the base period. The industrial structural component of b1 in the EM2 model is positive, and the specialization advantage based on the industrial scale in the base period is reflected. Therefore, shift-share EM1 and EM2 models can both realize the separation of the structural component and competitive component at the regional level, and only the EM2 model can realize the separation of the structural component and competitive component at the industrial level.

*3.2. Verification of the Separation Effect between Structural Component and Competitive Component*

The basic idea of shift-share analysis represented by the classical model and structural base model is that the shift component is obtained by subtracting the expected increment (share component) from the actual increment of the industry. Then, the shift component is further decomposed into a structural component and a competitive component according to the difference in industrial structure and growth rate. The essence of this kind of shift-share model is not the decomposition of the regional employment growth composition but the decomposition of the interregional employment growth deviation from balanced growth. However, the structural component and competitive component are intertwined in the shift-share model. Rosenfeld expressed this problem as follows: as the two regions have different employment in the same industry, even if the total employment in the region is the same and the industry grows at the same rate, its competitive component will be different [15]. To verify the separation effect of the EM2 model on the structural component and competitive component, the above hypothetical data are also used for analysis, and the calculation results are shown in Table 2.

The industrial employment increment is the product of the industrial scale in the base period and the industrial growth rate. The difference between the industrial scale and growth rate will affect the deviation in industrial employment growth. To separate the influence of the industrial scale and industrial growth rate on the employment growth deviation, the shift-share EM2 model first adopts the standard structure and average industrial growth rate to exclude the influence of industrial scale and growth rate difference in the share component so that regional share components with the same overall employment scale are equal and have the same distribution. Then, the structural component calculates the impact of the industrial structure difference in the base period on the employment growth deviation on the basis of equal industrial growth rate. The meaning of the structural

component is the net deviation scale of employment growth caused by the difference in the proportion of industrial scale (industrial structure/industrial specialization). The net deviation here means that only the influence of the industrial structure difference in the base period on growth deviation is considered on the basis of the same industrial growth rate. If the regional industrial structure in the base period is the same, then the structural components are equal. The competitive component calculates the impact of the industrial growth rate difference on the employment growth deviation on the basis of the same industrial structure in the base period. The meaning of the competitive component is the net deviation scale of employment growth caused by the difference in industrial growth rate (industrial competitiveness). The net deviation here means that only the influence of the industrial competitiveness difference on growth deviation is considered on the basis of the same industrial structure in the base period. If the regional industrial growth rate is the same, then the competitive components are equal. In this way, the structural and competitive components are completely separated. As seen in Table 2, region D and region A have the same industrial structure in the base period but different industrial growth rates. The structural components of the two regions are equal, both of which are 29.27, and the industrial distribution is the same, so the structural components are not affected by the difference in growth rates. The industrial growth rates of region C and region A are the same, but the industrial structure in the base period is different. The competitive components of the two regions are equal, both of which are −65.00, and the industrial distribution is the same. The competitive components are not affected by the industrial structure in the base period. Therefore, the shift-share EM2 model achieves the complete separation of the structural component and competitive component.

**Table 2.** Verification of the separation effect between the competitive component and structural component of the shift-share EM2 model based on the hypothetical data.

| Regions | Industries | $X_{ij}$ | $R_{ij}$ | $X'_{ij}R_i$ | $(X_{ij}-X'_{ij})R_i$ | $\sum_{i=1}^{s}(X_{ij}-X'_{ij})R_i$ | $X'_{ij}(R_{ij}-R_i)$ | $\sum_{i=1}^{s}X'_{ij}(R_{ij}-R_i)$ | $\sum_{i=1}^{s}(X_{ij}-X'_{ij})(R_{ij}-R_i)$ |
|---|---|---|---|---|---|---|---|---|---|
| | a1 | 200 | 0.2 | 72.50 | −19.77 | | −17.50 | | 4.77 |
| A | a2 | 500 | 0.5 | 205.00 | 68.33 | 29.27 | −17.50 | −65.00 | −5.83 |
| | a3 | 300 | 0.3 | 135.00 | −19.29 | | −30.00 | | 4.29 |
| | b1 | 500 | 0.3 | 72.50 | 59.32 | | 10.00 | | 8.18 |
| B | b2 | 200 | 0.6 | 205.00 | −95.67 | −55.63 | 20.00 | 70.00 | −9.33 |
| | b3 | 300 | 0.5 | 135.00 | −19.29 | | 40.00 | | −5.71 |
| | c1 | 200 | 0.2 | 72.50 | −19.77 | | −17.50 | | 4.77 |
| C | c2 | 300 | 0.5 | 205.00 | −41.00 | −2.92 | −17.50 | −65.00 | 3.50 |
| | c3 | 500 | 0.3 | 135.00 | 57.86 | | −30.00 | | −12.86 |
| | d1 | 200 | 0.3 | 72.50 | −19.77 | | 10.00 | | −2.73 |
| D | d2 | 500 | 0.6 | 205.00 | 68.33 | 29.27 | 20.00 | 70.00 | 6.67 |
| | d3 | 300 | 0.5 | 135.00 | −19.29 | | 40.00 | | −5.71 |

*3.3. Reinterpreting the Meaning of the Allocation Component*

In the allocation component, $(X_{ij} - X'_{ij})$ reflects the characteristic of industrial specialization. The proportion of the industry in the base period is larger than the average proportion of this industry in all regions, indicating that the industry in the base period has the advantage of specialization. In the allocation component, $(R_{ij} - R_i)$ reflects the characteristic of industrial competitiveness. The growth rate of the industry is faster than the average growth rate of all regions, indicating that the industry has competitive advantages. The meaning of the allocation component is explained by Esteban-Marquillas as follows: the allocation effect will be positive if the region is specialized in those sectors of faster regional growth or if it is not specialized in the sectors in which it is lacking in competitive advantages. In contrast, the allocation effect will be negative if the region is specialized in sectors in which the region is currently lacking in advantages or if it is not specialized in the sectors in which it has those competitive advantages. For a whole region, the allocation effect will be larger the better its employment is distributed among the different sectors according to their respective advantages [16]. The above description of the allocation component only reveals the distribution characteristic of the regional

industrial structure with relative competitive advantage. However, it does not directly explain the meaning of the allocation component.

To illustrate the meaning of allocation, we graphically outline four growth scenarios (see Figure 1). The industrial economic increment (cube C0) is the product of industrial scale (base area) and growth rate (height). The share component (cube C1) is the product of the industrial scale based on the standard structure and the average industrial growth rate.

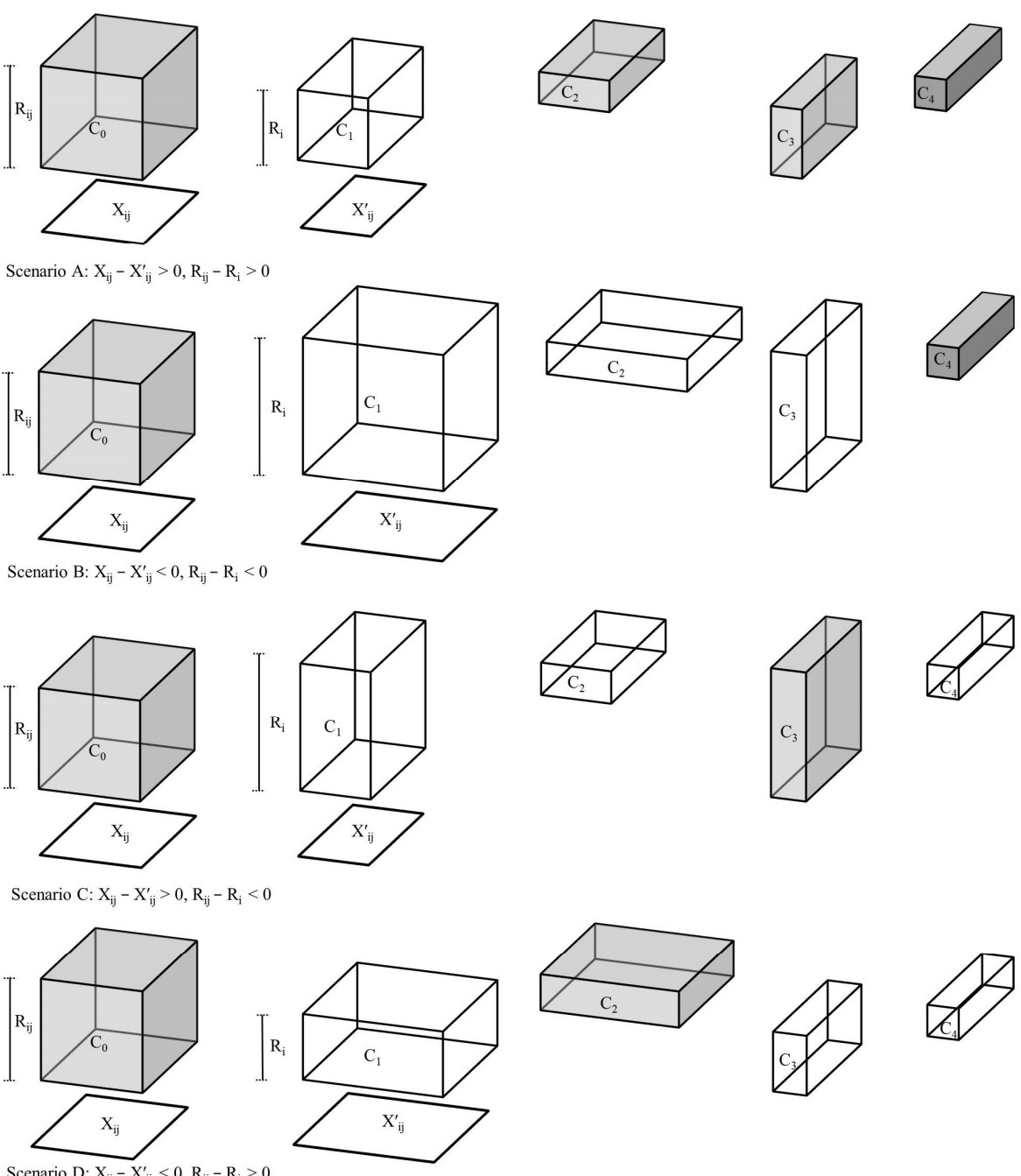

**Figure 1.** Component value combination types in four employment growth scenarios. Note: Cube C0 is the industrial employment increment, cube C1 is the share component, cube C2 is the structural component, cube C3 is the competitive component, cube C4 is the allocation component. For the structural component and competitive component, the gray cube represents the positive deviation, and the white cube represents the negative deviation.

The difference between the actual increment and the expected increment (share component) of the industry is the shift component, and the shift component is the sum of the structural component, competitive component, and allocation component. Thus, there are:

Shift component = actual increment − share component = structure component + competitive component + allocation component

Now, we introduce a new concept, the net shift component, which is defined as the increment generated by the initial industry scale with full reference to the standard structure and the average growth rate of the industry in all regions. Therefore, the net shift component is the difference between the shift component and the allocation component, that is, the allocation component is the difference between the shift component and the net shift component:

Net shift component = structural component + competitive component = shift component − allocation component

Allocation component = shift component − net shift component = shift component − (structural component + competitive component)

The structural component of the shift-share EM2 model calculates the impact of the industrial proportion difference on the employment growth deviation in the base period on the basis of equal industrial growth rate, which reflects the advantages (disadvantages) of initial industrial specialization. On the basis of the same industrial proportion in the base period, the competitive component calculates the impact of industrial growth rate differences on employment growth deviation, which reflects the competitive advantages (disadvantages) of industries. Thus, the meaning of the allocation component reflects the inhibition or magnification effect of the competitive component on the structural component in the process of employment growth during the study period.

In scenario A, the industrial initial scale and growth rate are both higher than the expectation of balanced growth, and the sum of the structural component and competitive component is less than the industrial shift component. The meaning of the allocation component in this scenario is the magnification effect of the industrial competitive advantage on the initial structural advantage of the industry in the process of growth. In scenario B, the industrial scale and growth rate are both lower than the expectation of balanced growth, and the sum of the structural component and competitive component is less than the industrial shift component. The meaning of the allocation component in this scenario is the magnification effect of the industrial competitive disadvantage on the initial structural disadvantage of the industry in the process of growth. In scenario C, the industrial scale is higher than the expectation of balanced growth, but the growth rate is lower than the expectation of balanced growth, and the sum of the structural component and the competitive component is greater than the industrial shift component. The meaning of the allocation component in this scenario is the inhibition effect of the industrial competitive disadvantage on the initial structural advantage of the industry in the process of growth. In scenario D, the industrial scale is lower than the expectation of balanced growth, while the growth rate is higher than the expectation of balanced growth, and the sum of the structural component and competitive component is greater than the industrial shift component. The meaning of the allocation component in this scenario is the inhibition effect of the initial structural disadvantage of the industry on the industrial competitive advantage in the process of growth.

## 4. Application of the Shift-Share EM2 Model

In order to apply the shift-share EM2 model in the analysis of regional employment growth, an empirical study is conducted based on the manufacturing employment data of China from 1999 to 2019. The manufacturing industry has laid a solid foundation for China's economic development. The technological upgrading and structural adjustment of the manufacturing industry have also deeply affected the scale and pattern of employment in China. The bursting of the internet bubble in 2000 led to more investment in the real economy. China's accession to the World Trade Organization in 2001 further promoted

the process of specialization in China's manufacturing industry. The period from 1999 to 2019, before the full outbreak of COVID-19, was a golden period of rapid development of China's manufacturing industry. The structure of the manufacturing industry changed significantly and the trend of unbalanced development was obvious. Data sources are the China Industrial Statistical Yearbook 2000 and the China Industrial Statistical Yearbook 2020. To facilitate the calculation of the structural component, the manufacturing industry is grouped into six categories (In the China Industrial Statistical Yearbook 2000, employment data of manufacturing industries such as wood, furniture, paper making, printing, and stationery are not counted. In order to ensure consistent industrial grouping before and after, industries that are not counted are excluded. The six manufacturing sectors involved in the summary calculation are as follows: Food: agricultural and sideline food processing, food manufacturing, wine, beverage and refined tea manufacturing, and tobacco manufacturing. Garments: Textiles, textile garments, apparel, leather, fur, feather and their products, and footwear. Chemical: petroleum processing, coking and nuclear fuel processing, chemical raw materials and chemical products manufacturing, pharmaceutical manufacturing, chemical fiber manufacturing, rubber and plastic products industry. Metallurgy: non-metallic mineral products, ferrous metal smelting and calendering processing industry, non-ferrous metal smelting and calendering processing industry, metal products industry. Equipment: General equipment manufacturing, special equipment manufacturing, automobile manufacturing, railway, shipbuilding, aerospace, and other transportation equipment manufacturing. Electronics: Electrical machinery and equipment manufacturing, computer, communication and other electronic equipment manufacturing, instrument manufacturing): food, garments, chemical, metallurgy, equipment, and electronics.

### 4.1. Application at the Regional Level

The regional shift component is the sum of all industrial competitive components, structural components, and allocation components. It is also the gap between the actual increment and the expected growth of the region. The regional shift component reflects the deviation scale of regional employment growth. Table 3 shows the calculation results of the regional shift component of China's manufacturing industry from 1999 to 2019. The results show that the spatial imbalance trend of employment growth in China's manufacturing industry is significant during the study period, and the deviation scale of employment growth forms a north–south difference pattern. Specifically, eight provinces and cities, Liaoning, Heilongjiang, Jilin, Beijing, Tianjin, Shanxi, Gansu, and Xizang, record negative growth in the manufacturing employment during the study period. Only 8 of the 31 regions have positive regional shift components, while the remaining have negative regional shift components. The eight provinces and cities with positive deviations are distributed south of the Yangtze River, among which Guangdong, Fujian, and Zhejiang on the southeast coast account for 82% of the positive deviation. It is worth noting that 96% of the positive deviation in employment growth in the three regions is due to the competitive component generated by regional competitiveness. The highest deviation in employment growth in Guangdong is 6,037,700 people, accounting for approximately half of the positive deviation. The deviation in employment growth shows a strong trend of agglomeration. Provinces and cities with a large scale of negative deviation in employment growth mainly appear in Northeast and North China, accounting for approximately 60% of the total negative deviation.

Due to the difference in employment scale, the degree of regional deviation cannot be directly compared with the regional shift component. Taking Fujian as an example, its regional shift component is only 38% of that of Guangdong. However, compared with the share component of 457,700 people, the deviation degree is far higher than that of Guangdong. To compare the deviation degree of each region, we put forward the parameter

$S_j$ (regional deviation degree), which can be calculated as follows:

$$S_j = \frac{\sum\limits_{i=1}^{m} X_{ij}R_{ij} - \sum\limits_{i=1}^{m} X'_{ij}R_i}{\left|\sum\limits_{i=1}^{m} X'_{ij}R_i\right|} \quad (12)$$

**Table 3.** The deviation scale of employment growth in China's manufacturing industry from 1999 to 2019 based on the shift-share EM2 model.

| Regions | Increment | Share Component | Structural Component | Competitive Component | Allocation Component | Shift Component |
|---|---|---|---|---|---|---|
| Guangdong | 793.91 | 190.14 | 128.42 | 587.74 | −112.39 | 603.77 |
| Fujian | 275.46 | 45.77 | 9.54 | 251.92 | −31.77 | 229.69 |
| Zhejiang | 331.41 | 133.39 | 25.55 | 154.51 | 17.97 | 198.02 |
| Jiangxi | 112.66 | 49.68 | −6.88 | 81.62 | −11.76 | 62.98 |
| Anhui | 117.09 | 66.28 | −9.25 | 83.42 | −23.36 | 50.81 |
| Jiangsu | 310.26 | 261.48 | 35.39 | 15.78 | −2.38 | 48.78 |
| Hunan | 122.81 | 74.50 | −15.32 | 72.73 | −9.09 | 48.31 |
| Chongqing | 59.68 | 41.61 | −5.10 | 22.47 | 0.70 | 18.07 |
| Xizang | −0.13 | 0.63 | −0.33 | −0.59 | 0.15 | −0.76 |
| Henan | 140.58 | 143.37 | −26.23 | 39.57 | −16.12 | −2.79 |
| Ningxia | 5.43 | 8.55 | −2.32 | −3.09 | 2.29 | −3.12 |
| Hainan | 0.25 | 4.47 | −0.83 | −5.58 | 2.20 | −4.22 |
| Qinghai | 0.24 | 7.04 | −2.94 | −6.19 | 2.33 | −6.80 |
| Xinjiang | 11.40 | 19.77 | −4.28 | −6.68 | 2.59 | −8.37 |
| Neimenggu | 8.79 | 26.34 | −10.01 | −23.41 | 15.87 | −17.55 |
| Guangxi | 19.53 | 42.25 | −7.38 | −18.73 | 3.39 | −22.72 |
| Yunnan | 5.14 | 31.08 | −9.98 | −31.57 | 15.61 | −25.94 |
| Sichuan | 66.69 | 95.06 | −6.45 | −24.73 | 2.81 | −28.37 |
| Guizhou | 0.82 | 29.47 | −6.87 | −25.91 | 4.12 | −28.65 |
| Hubei | 87.28 | 119.14 | −20.07 | −16.73 | 4.94 | −31.86 |
| Shaanxi | 18.36 | 56.45 | 9.43 | −33.35 | −14.17 | −38.09 |
| Tianjin | −9.40 | 53.50 | 8.38 | −67.57 | −3.71 | −62.90 |
| Gansu | −29.67 | 36.87 | −10.32 | −69.21 | 12.99 | −66.54 |
| Shanxi | −20.18 | 60.73 | −23.47 | −63.60 | 6.16 | −80.91 |
| Jilin | −28.93 | 56.27 | −9.08 | −92.02 | 15.89 | −85.20 |
| Beijing | −28.87 | 57.21 | 8.01 | −90.78 | −3.31 | −86.08 |
| Hebei | 21.47 | 115.32 | −33.19 | −80.97 | 20.31 | −93.85 |
| Shanghai | 1.95 | 102.54 | 22.21 | −117.88 | −4.93 | −100.59 |
| Heilongjiang | −55.89 | 56.84 | −2.65 | −116.18 | 6.10 | −112.73 |
| Shandong | 78.57 | 212.05 | −14.86 | −123.39 | 4.76 | −133.48 |
| Liaoning | −84.23 | 134.66 | −19.12 | −215.56 | 15.79 | −218.89 |

Here, the numerator is the gap between the regional employment increment and the share component, i.e., the regional shift component; the denominator is the share component. To avoid the positive value of the regional deviation degree when both the regional shift component and the share component are negative at the same time, the absolute value of the share component is taken. Obviously, when the employment increment is exactly equal to the share component, the regional deviation degree $S_j$ is equal to 0. That means the regional employment growth just achieves the expected growth. Based on the deviation degree, the range of the regional deviation degree can be divided into mild positive deviation (0, 0.2), moderate positive deviation (0.2, 0.5), high positive deviation (0.5, 1), and extreme positive deviation (1, +∞). In the same way, the interval of the negative deviation is the same, but the sign is the opposite. The classification results are shown in Figure 2.

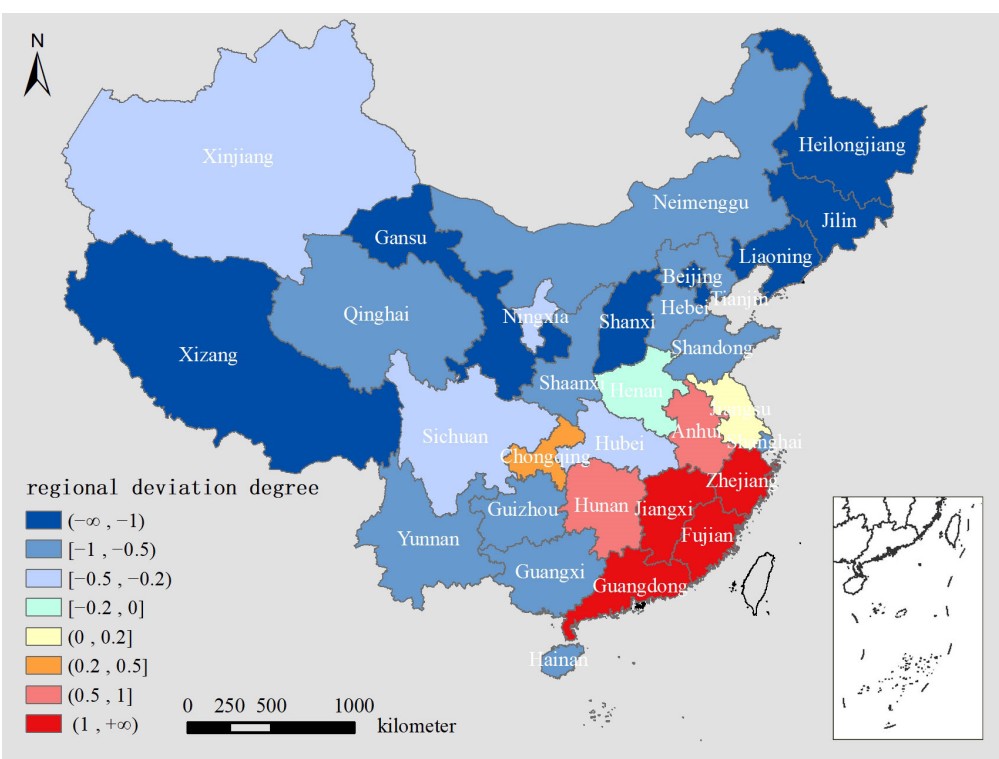

**Figure 2.** The regional deviation degree pattern of employment growth in China's manufacturing industry from 1999 to 2019 based on the shift-share EM2 model. Note: No data are available for Hong Kong, Macao, and Taiwan.

The spatial pattern of regional deviation degree is southeast to northwest. Among the eight positive deviation regions, Fujian, Guangdong, Zhejiang, and Jiangxi Provinces have extremely positive deviations. The expected value (share component) of Fujian's manufacturing employment growth is 457,700, while the deviation is 2,696,900, with the highest regional deviation degree of 5.02. Guangdong, which ranks second, also has a regional deviation degree of 3.18. The region with an extreme deviation degree of employment growth is mainly distributed in the southeast coastal areas. On the one hand, the continuous improvement of transportation and information technology has created conditions for the transfer-out of the surplus rural labor force. On the other hand, in the process of reform and opening up, southeast coastal areas give priority to the development of export-oriented labor-intensive manufacturing industry. Hunan and Anhui have high positive deviations. Chongqing and Jiangsu have moderate positive deviations and slight positive deviations, respectively. The scale of regional deviation in Jiangsu and Hunan is very similar, 487,800 and 483,100, respectively. However, the share component of Jiangsu as an expected growth is 3.1026 million, while that of Hunan is only 1.228 million. Therefore, the regional deviation degree of Hunan is high positive deviation, while that of Jiangsu is only moderate positive deviation. Compared with the scale of regional deviation, the regional deviation degree can better reflect the deviation degree of regional employment unbalanced growth. The eight provinces and cities with extreme negative deviations are Heilongjiang, Gansu, Liaoning, Jilin, Beijing, Shanxi, Xizang, and Tianjin. There is a slight negative deviation in Henan Province, and a moderate negative deviation in Sichuan, Hubei, Ningxia, and Xinjiang Provinces. The remaining ten provinces and cities with a high negative deviation are mainly distributed in North China and Northwest and Southwest China.

### 4.2. Application at the Industrial Level

To demonstrate the application of the shift-share EM2 model at the industrial level, Sichuan Province, which contains all four component combination types, is selected as an example, see Table 4. The structural and competitive components of the Sichuan food manufacturing industry are both positive during the study period, and the industrial initial scale and growth rate are both higher than the expectation of balanced growth. The positive net deviation scale of employment growth caused by the industrial structure difference is 0.17 million, and the positive net deviation scale of employment growth caused by the industrial competitiveness difference is 17.93 million. The food manufacturing industry has both a structural advantage and competitive advantage. In this scenario, the meaning of the allocation component is the magnification effect of the industrial competitive advantage on the industrial initial structural advantage.

The structural and competitive components of garments, electronics, and metallurgical manufacturing are both negative, and the industrial scale and growth rate are both lower than the expectation of balanced growth. The negative net deviation scale of the employment growth of the garments, metallurgical, and electronics manufacturing industry caused by the industrial structure difference is 3.75, 0.07, and 1.51 million, respectively, and the negative net deviation scale of employment growth caused by the industrial competitiveness difference is 17.13, 6.36, and 12.06 million, respectively. The garments, metallurgical, and electronics manufacturing industry have neither structural nor competitive advantages. In this scenario, the meaning of the allocation component is the magnification effect of the industrial competitive disadvantage on the initial structural disadvantage of the industry in the process of growth.

The structural component of the equipment manufacturing industry is positive, and the competitive component is negative. The industrial scale is higher than the expectation of balanced growth, but the growth rate is lower than the expectation of balanced growth. The positive net deviation scale of employment growth of the equipment manufacturing industry caused by the industrial structure difference is 0.44 million, and the negative net deviation scale of employment growth caused by the industrial competitiveness difference is 12.42 million. The equipment manufacturing industry has a structural advantage but no competitive advantage. In this scenario, the meaning of the allocation component is the inhibition effect of the industrial competitive disadvantage on the initial structural advantage of the industry in the process of growth.

The structural component of the chemical manufacturing industry is negative, while the competitive component is positive. The industrial scale is lower than the expectation of balanced growth, while the growth rate is higher than the expectation of balanced growth. The negative net deviation scale of employment growth of the chemical manufacturing industry caused by the industrial structure difference is 1.73 million, and the positive net deviation scale of employment growth caused by the industrial competitiveness difference is 5.71 million. The chemical manufacturing industry has a competitive advantage but no structural advantage. In this scenario, the meaning of the allocation component is the inhibition effect of the initial structural disadvantage of the industry on the industrial competitive advantage in the process of growth.

The application of the shift-share EM2 model at the industrial level can be the reference for the analysis of industrial structure. When the structural and competitive components are both positive, the industrial competitive advantage strengthens the initial structural advantage, and this industry can be considered as a leading industry. When the structural and competitive components are both negative, the industrial competitive disadvantage strengthens the initial structural disadvantage, and this industry can be considered as a sunset industry. When the structural component is positive and the competitive component is negative, the industrial competitive disadvantage inhibits the initial structural advantage, and the industry can be considered as a pillar industry if it has a certain scale. When the structural component is negative and the competitive component is positive, the initial

structural disadvantage inhibits the industrial competitive advantage, and this industry can be considered as a latent guiding industry.

**Table 4.** The component distribution forms of employment growth in Sichuan Province's manufacturing industry from 1999 to 2019 based on the shift-share EM2 model.

| Industries | Base Scale | End Scale | Growth Rate | Share Component | Shift Component | Structural Component | Competitive Component | Allocation Component |
|---|---|---|---|---|---|---|---|---|
| food | 17.14 | 43.3 | 1.53 | 7.68 | 18.48 | 0.17 | 17.93 | 0.39 |
| garments | 15.35 | 13.25 | −0.14 | 14.28 | −16.38 | −3.75 | −17.13 | 4.50 |
| chemical | 23.69 | 38.28 | 0.62 | 11.79 | 2.80 | −1.73 | 5.31 | −0.78 |
| metallurgy | 49.22 | 41.04 | −0.17 | −0.31 | −7.87 | −0.07 | −6.36 | −1.44 |
| equipment | 34.96 | 41.67 | 0.19 | 18.97 | −12.26 | 0.44 | −12.42 | −0.29 |
| electronic | 18.58 | 48.09 | 1.59 | 42.65 | −13.14 | −1.51 | −12.06 | 0.43 |

## 5. Conclusions

The essence of the shift-share model, which is based on the classical model and structural base model, is not the decomposition of the composition of regional employment growth but the decomposition of the interregional employment growth deviation from balanced growth. The shift-share model uses the share component as the reference standard to represent the balanced growth of regional employment. The balanced growth of the share component set by the EM2 model means that the industrial structure in the base period is the same in all regions, and the growth rate of the same industry in all regions is the same. The regions with the same overall employment scale have the same share components and the industrial structure. The share component of the EM1 model is set as all industries and regions grow at the same rate, which means that the industrial structures of all regions remain unchanged but different from each other in the growth process. The unbalanced distribution state of the base period remains unchanged. The share and structural components of the EM1 and EM2 models are equal according to the sum of industries, but the distribution is different among industries. The structural component of the EM2 model can better reveal the advantage of specialization of regional employment growth in the base period.

Esteban-Marquillas's explanation of the allocation component just reveals the distribution characteristics of the regional industrial structure with relative competitive advantage, without directly explaining the meaning of the allocation component. In fact, the meaning of the allocation component of the EM2 model reflects the inhibition or magnification effect of the industrial competitive advantage (or disadvantage) on the initial structural advantage (or disadvantage) in the process of employment growth during the study period.

Due to the difference in employment scale in different regions, the degree of employment deviation in different regions cannot be directly compared with the shift component. We propose the parameter Sj (regional deviation degree) to compare the degree of regional employment growth deviation.

The application study analyzes the employment growth of China's manufacturing industry from 1999 to 2019 based on the employment data of different sectors in China's manufacturing industry. The results show that the spatial imbalance trend of employment growth in China's manufacturing industry is obvious, the scale of employment growth deviation forms a north–south difference pattern, and the spatial pattern of the degree of deviation tends to be southeast to northwest. Taking Sichuan Province as an example, the application of the shift-share EM2 model at the industrial level and the meaning of the structural component, competitive component, and allocation component under different situations are demonstrated. The application of the shift-share EM2 model at the industrial level can be the reference for the analysis of industrial structure.

## 6. Discussion

To separate structural and competitive components, Esteban-Marquillas introduced the standard structure into the classical model and structural base model, respectively, to form the EM1 and EM2 models. In the shift-share EM2 model, the standard structure and average industrial growth rate are adopted to exclude the influence of scale and growth rate differences on the share component. Then, the structural component calculates the impact of the industrial structure difference on the employment growth deviation in the base period on the basis of equal industrial growth rate. On the basis of the same industrial structure in the base period, the competitive component calculates the impact of the industrial growth rate difference on the employment growth deviation. In this way, the shift-share EM2 model realizes the separation of the employment growth deviation caused by the industrial structure and industrial competitiveness at the industrial level.

If the structural component and the competitive component are calculated based on the actual scale and growth rate of the industry, such a calculation would inevitably interweave the structural component and the competitive component. The structural component and competitive component calculated by the average industrial growth rate and the industrial scale treated by the standard structure are the net shift component. The difference between the shift component and the net shift component is the allocation component. Combining the positive and negative combination types of the structural component, competitive component, and allocation component, we can judge whether the competitive advantage (or disadvantage) strengthens the structural advantage (disadvantage), or the competitive disadvantage inhibits the structural advantage, or the structural disadvantage inhibits the competitive advantage, at the industrial level.

In the future, on the one hand, dynamic models can be used to analyze the evolution process of different regions and industry deviation types. On the other hand, we can compare the difference between the analysis results of employment data and output value data for the case with a long time interval.

**Author Contributions:** Conceptualization, J.L.; Methodology, J.L.; Writing—original draft, J.L.; Funding acquisition, Y.Y. All authors have read and agreed to the published version of the manuscript.

**Funding:** This research was funded by the National Natural Science Foundation of China, Grant Number: 41971198.

**Institutional Review Board Statement:** Not applicable.

**Informed Consent Statement:** Not applicable.

**Data Availability Statement:** The data for this research can be obtained from J.L. upon reasonable request.

**Conflicts of Interest:** The authors declare no conflict of interest.

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
