# Peer review of "Rediscovering the Shift-Share EM2 Model: A Decomposition Framework of Unbalanced Employment Growth at the Industrial Level"

_sustainability, doi:10.3390/su15065039_

Round 1

Reviewer 1 Report

This is an interesting study that aims to rediscover the shift-share EM2 model which can provide a decomposition framework of unbalanced employment growth at the industrial level. The differences between the shift-share EM1 and EM2 models are compared. Finally, the shift-share EM2 model is applied in analyzing regional unbalanced employment growth by using manufacturing employment data from China in 1999 and 2019. Below, some comments that may improve the quality of the manuscript and some confusions.

  (1) In the part of Abstract the main research results need to be further presented, and now it is only about what kind of work has been done, and the main results cannot be seen.

  (2) I think a full literature review is needed. From your paper, I can't see the innovation and theoretical contribution of your paper very well.

(3) In the empirical part, what is the effect of the method, and the comparison with the traditional method needs to be strengthened.

(4) Finally, a discussion section is needed, discussing the scope and effects of the method.

Author Response

Dear reviewer,

Thank you very much for your support and valuable comments. According to your suggestion, the paper is revised as follows:

This is an interesting study that aims to rediscover the shift-share EM2 model which can provide a decomposition framework of unbalanced employment growth at the industrial level. The differences between the shift-share EM1 and EM2 models are compared. Finally, the shift-share EM2 model is applied in analyzing regional unbalanced employment growth by using manufacturing employment data from China in 1999 and 2019. Below, some comments that may improve the quality of the manuscript and some confusions.

  (1) In the part of Abstract the main research results need to be further presented, and now it is only about what kind of work has been done, and the main results cannot be seen.

Therefore, shift-share EM2 model can realize the separation of structural component and competitive component at the industrial level.

The results show that the spatial imbalance trend of employment growth in China's manufacturing industry is significant, the regional deviation scale of employment growth forms a north-south difference pattern, and the regional deviation degree forms a southeast to northwest difference pattern.

  (2) I think a full literature review is needed. From your paper, I can't see the innovation and theoretical contribution of your paper very well.

I am very sorry that the application of shift-share EM model is basically used in the EM1 model. As for the research on EM2 model, Keil pointed out that EM2 and EM1 models are equal at the regional level. The differences between the two models at the industrial level have been pointed out in Section 3.1. In another important literature, Loveridge acknowledges the separation effect of the EM2 model, but believes that the EM2 model has no practical reference value. In fact, the cognition of the EM2 model r requires reinterpretation to the allocation component and figure out its relation to the net shift component. I'll add Loveridge's point to the literature review. In addition, the internal relationship of each component in EM2 model is discussed in detail in Section 3.3.

The difference between the actual increment and the expected increment (share component) of the industry is the shift component, and the shift component is the sum of the structural component, competitive component and allocation component. So there are:

Shift component = actual increment - share component = structure component + competitive component + allocation component

Now, we introduce a new concept, the net shift component, which is defined as the increment generated by the initial industry scale with full reference to the standard structure and the average growth rate of the industry in all region. Therefore, the net shift component is the difference between the shift component and the allocation component, that is, the allocation component is the difference between the shift component and the net shift component:

Net shift component = structural component + competitive component = shift component - allocation component

Allocation component = shift component - net shift component = shift component - (structural component + competitive component)

 (3) In the empirical part, what is the effect of the method, and the comparison with the traditional method needs to be strengthened.

 The scale of regional deviation in Jiangsu and Hunan is very similar, 487,800 and 483,100, respectively. However, the share component of Jiangsu as an expected growth is 3.1026 million, while that of Hunan is only 1.228 million. Therefore, the regional deviation degree of Hunan is high positive deviation, while that of Jiangsu is only moderate positive deviation. Compared with the scale of regional deviation, regional deviation degree can better reflect the deviation degree of regional employment unbalanced growth.

The differences and advantages of EM2 model compared with EM1 model and classical model in industrial level need to be explained by strictly assumed data. Appropriate empirical cases are hard to find. Therefore, it is emphasized again in section 3.1:

Therefore, shift-share EM1 and EM2 models both can realize the separation of structural component and competitive component at the regional level, only the EM2 model can realize the separation of structural component and competitive component at the industrial level.

(4) Finally, a discussion section is needed, discussing the scope and effects of the method.

  1. 6. DISCUSSION

To separate structural and competitive components, Esteban-Marquillas introduced the standard structure into the classical model and structural base model, respectively, to form the EM1 and EM2 models. In the shift-share EM2 model, the standard structure and average industrial growth rate are adopted to exclude the influence of scale and growth rate differences on the share component. Then, the structural component calculates the impact of industrial structure difference on employment growth deviation in the base period on the basis of equal industrial growth rate. On the basis of the same industrial structure in the base period, the competitive component calculates the impact of the industrial growth rate difference on employment growth deviation. In this way, the shift-share EM2 model realized the separation of the employment growth deviation caused by industrial structure and industrial competitiveness at the industrial level.

If the structural component and the competitive component are calculated based on the actual scale and growth rate of the industry, such a calculation would inevitably interweave the structural component and the competitive component. The structural component and competitive component calculated by the average industrial growth rate and the industrial scale treated by the standard structure are the net shift component. The difference between the shift component and the net shift component is the allocation component. Combining the positive and negative combination types of structural component, competitive component and allocation component, we can judge whether the competitive advantage (or disadvantage) strengthens the structural advantage (disadvantage), or the competitive disadvantage inhibits the structural advantage, or the structural disadvantage inhibits the competitive advantage, at the industrial level.

Reviewer 2 Report

I have read the paper with interest and I think it is suitable for publication by Sustainability. 

Minor comments: 

1) Shift and Share should be better conceptually explained in the first section. Before introducing all the improvements/differences proposed by different scholars the Shift Share should be thoroughly explained.

2) The focus on manufacturing should be better explained and justified

3) Some more policy-related conclusions are missing as well as some ideas for further research stemming from the obtained results. Apart from being an application of the methodology to China, which is the main messages that emerge?

Author Response

Dear reviewer,

Thank you very much for your support and valuable comments. According to your suggestion, the paper is revised as follows:

I have read the paper with interest and I think it is suitable for publication by Sustainability. 

Minor comments: 

1) Shift and Share should be better conceptually explained in the first section. Before introducing all the improvements/differences proposed by different scholars the Shift Share should be thoroughly explained.

The basic methodology of shift-share analysis is deviation analysis and accounting decomposition structure. First, the employment growth is decomposed into the expected increment under the assumption of balanced growth (share component) and the growth deviation between the actual increment and the expected increment (shift component). Secondly, shift component is further decomposed into the structural component and competitive component which represent unbalanced growth caused by regional industrial structure and competitiveness differences.

2) The focus on manufacturing should be better explained and justified

The manufacturing industry has laid a solid foundation for China's economic development. The technological upgrading and structural adjustment of the manufacturing industry have also deeply affected the scale and pattern of employment in China.

The structure of the manufacturing industry changed significantly and the trend of unbalanced development was obvious.

3) Some more policy-related conclusions are missing as well as some ideas for further research stemming from the obtained results. Apart from being an application of the methodology to China, which is the main messages that emerge?

The application of shift-share EM2 model at the industrial level can be the reference for the analysis of industrial structure. When the structural and competitive components are both positive,industrial competitive advantage strengthens the initial structural advantage, this industry can be considered as a leading industry. When the structural and competitive components are both negative, industrial competitive disadvantage strengthens initial structural disadvantage, this industry can be considered as a sunset industry. When the structural component is positive and the competitive component is negative, industrial competitive disadvantage inhibits the initial structural advantage, the industry can be considered as a pillar industry if it has a certain scale. When the structural component is negative and the competitive component is positive, the initial structural disadvantage inhibits the industrial competitive advantage, this industry can be considered as a latent guiding industry.

In the future, on the one hand, dynamic models can be used to analyze the evolution process of different regions and industry deviation types. On the other hand, we can compare the difference between the analysis results of employment data and output value data for the case with a long time interval.

Reviewer 3 Report

The article is well developed and also coherent with the research hypothesis.  

Abstract:

The abstract is , focuses on the research question, the methodology applied and offers some first findings.

Introduction:

The introduction, starts with a literature analysis and the applied model is separately considered and well described.

Methodology: is explainedcated.

Conclusion:

The case of Sichuan Province is a good example to check the model components and results.

1. What is the main question addressed by the research?   The research question wants to demonstrate the relationship between unbalanced employment growth among regions attributed to the difference in industrial structure and industrial competitiveness.  This is a long debate in literature and some regional development models. In this way, you may consider the clusters theory and the Industrial district from Marshall theory. He affirms that concentration is also connected with an efficient labour market.    2. Do you consider the topic original or relevant in the field? Does it  address a specific gap in the field?   Industrial policies, particularly in the manufacturing sector, and the spontaneous or planned concentration of industries are interesting to study, particularly in China. The expanding economy of China needs a plan not only considering the industry concentration, generally spatial related to the market, but also a labour policy shod be considered by the Government. This article explores public policy in China, connecting industrial development with labour policies.      3. What does it add to the subject area compared with other published  material? The article adds a competitive aspect to the localization industry process. The literature explored those relations in EU countries. So the new aspect is the competitive aspect, which could also be considered in some provinces in China.      4. What specific improvements should the authors consider regarding the  methodology? What should further controls be considered? The EM1 and EM2 models were applied and could better specify the adoption of EM2.    5. Are the conclusions consistent with the evidence and arguments presented and do they address the main question posed? The article's findings are also appropriate if based on a specific Chinese province.    6. Do you know if the references are appropriate? I think references are appropriate, and the authors can also consider the Industrial District model for the literature discussion.    7. Please include any additional comments on the tables and figures.

Figure 1 could have a higher reader impact and find a raffiguration more easily. 

Author Response

Dear reviewer,

Thank you very much for your support and valuable comments. According to your suggestion, the paper is revised as follows:

The article is well developed and also coherent with the research hypothesis.  

Abstract:

The abstract is , focuses on the research question, the methodology applied and offers some first findings.

Introduction:

The introduction, starts with a literature analysis and the applied model is separately considered and well described.

Methodology: is explainedcated.

Conclusion:

The case of Sichuan Province is a good example to check the model components and results.

  1. What is the main question addressed by the research?   The research question wants to demonstrate the relationship between unbalanced employment growth among regions attributed to the difference in industrial structure and industrial competitiveness.  This is a long debate in literature and some regional development models. In this way, you may consider the clusters theory and the Industrial district from Marshall theory. He affirms that concentration is also connected with an efficient labour market.

The systematic research on industrial agglomeration started from Marshall's industrial district theory, and he believed that it was the external economy composed of specialized input, centralized job market and knowledge spillover that led to industrial spatial agglomeration (Marshall, 1890). The new industrial district theory, which takes "Third Italy", a social regional production complex dominated by small and medium-sized enterprises, as the research object, holds that the organization of external transaction costs based on competitive cooperative relationship enterprise network is the basis of industrial agglomeration. (Bagnasco, 1977; Scott, 1988). New economic geography assumes increasing returns to scale and imperfect competitive markets, and holds that increasing returns to scale, reducing transport costs and flow of production factors promote industrial agglomeration through market effects (Krugman, 1991). The regional innovation system theory emphasizes the importance of socio-cultural background and spatial proximity to high-tech industry agglomeration (Cooke, 1993). According to the social capital theory, economic behavior is embedded in the social relationship network and has the embeddedness to deviate from the goal of benefit maximization (Granoetter, 1985).

  1. Do you consider the topic original or relevant in the field? Does it address a specific gap in the field?   Industrial policies, particularly in the manufacturing sector, and the spontaneous or planned concentration of industries are interesting to study, particularly in China. The expanding economy of China needs a plan not only considering the industry concentration, generally spatial related to the market, but also a labour policy shod be considered by the Government. This article explores public policy in China, connecting industrial development with labour policies.     

The region with high regional deviation degree of employment growth is mainly distributed in the southeast coastal areas. On the one hand, the continuous improvement of transportation and information technology has created conditions for the transfer-out of surplus rural labor force. On the other hand, in the process of reform and opening up, southeast coastal areas give priority to the development of export-oriented labor-intensive manufacturing industry.

  1. What does it add to the subject area compared with other published material? The article adds a competitive aspect to the localization industry process. The literature explored those relations in EU countries. So the new aspect is the competitive aspect, which could also be considered in some provinces in China.     

It is worth noting that 96% of the positive deviation of employment growth in the three regions is due to competitive component generated by regional competitiveness

  1. What specific improvements should the authors consider regarding the  methodology? What should further controls be considered? The EM1 and EM2 models were applied and could better specify the adoption of EM2.   

Therefore, shift-share EM1 and EM2 models both can realize the separation of structural component and competitive component at the regional level, only the EM2 model can realize the separation of structural component and competitive component at the industrial level.

  1. Are the conclusions consistent with the evidence and arguments presented and do they address the main question posed? The article's findings are also appropriate if based on a specific Chinese province.   

Indeed, if we want to analyze the unbalanced growth of manufacturing employment at the industry level, Guangdong and Fujian are both better cases. The author chose Sichuan Province as the case because the manufacturing industry in Sichuan Province contains all the deviation types, and the purpose is just to demonstrate the application of EM2 model at the industry level and the explanation of various industry types.

  1. Do you know if the references are appropriate? I think references are appropriate, and the authors can also consider the Industrial District model for the literature discussion.   

Bagnasco A, Cucchi P, Jalla E. Organization terrible dell industrial manufacturing in Italia.Fondazione Agnell, 1977.

Marshall A. Principles of economics. Macmillan,London,1890.

Scott A J. New industrial space.London:Pion,1988

Krugman P R, increasing returns and economic geography [J]. Journal of political economy. 1991,99(3):483-499.

Cooke P. regional innovation system, an evaluation of six European case. In Urban development and regional development in the new Europe. Athens:Topos, 1993,133-145.

Granoetter M. Economic action and social structure: the problem of embeddedness[J].The American journal of sociology,1985,91(3):481-510.

  1. Please include any additional comments on the tables and figures.

Figure 1 could have a higher reader impact and find a configuration more easily.

Note: Cube C0 is the industrial employment increment, cube C1 is the share component, cube C2 is the structural component, cube C3 is the competitive component, cube C4 is the allocation component. For the structural component and competitive component, the gray cube represents the positive deviation, and the white cube represents the negative deviation.
